# The Immunomodulatory Function of Vitamin D, with Particular Reference to SARS-CoV-2

**DOI:** 10.3390/medicina57121321

**Published:** 2021-12-02

**Authors:** Alberto Caballero-García, David C. Noriega, Hugo J. Bello, Enrique Roche, Alfredo Córdova-Martínez

**Affiliations:** 1Department of Anatomy and Radiology, Health Sciences Faculty, GIR of Physical Exercise and Aging, Campus Universitario “Los Pajaritos”, 42004 Soria, Spain; alberto.caballero@uva.es; 2Spine Department, Valladolid University Hospital, University of Valladolid, 47005 Valladolid, Spain; noriega1970@icloud.com; 3Department of Mathematics, School of Forestry Industry and Agronomic Engineering and Bioenergy, GIR of Physical Exercise and Aging, Campus Universitario “Los Pajaritos”, 42004 Soria, Spain; hjbello.wk@gmail.com; 4Department of Applied Biology-Nutrition, Institute of Bioengineering, University Miguel Hernández, 03202 Elche, Spain; eroche@umh.es; 5CIBER Physiopathology of Obesity and Nutrition (CIBEROBN), Instituto de Salud Carlos III (ISCIII), 28029 Madrid, Spain; 6Department of Biochemistry, Molecular Biology and Physiology, Faculty of Health Sciences, GIR of Physical Exercise and Aging, Campus Universitario “Los Pajaritos”, Valladolid University, 42004 Soria, Spain

**Keywords:** immunomodulation, vaccination, vitamin D

## Abstract

Vaccines are the only way to reduce the morbidity associated to SARS-CoV-2 infection. The appearance of new mutations urges us to increase the effectiveness of vaccines as a complementary alternative. In this context, the use of adjuvant strategies has improved the effectiveness of different vaccines against virus infections such as dengue, influenza, and common cold. Recent reports on patients infected by COVID-19 reveal that low levels of circulating vitamin D correlate with a severe respiratory insufficiency. The immunomodulatory activity of this micronutrient attenuates the synthesis of pro-inflammatory cytokines and at the same time, increases antibody production. Therefore, the present review proposes the use of vitamin D as adjuvant micronutrient to increase the efficacy of vaccines against SARS-CoV-2 infection.

## 1. Introduction

Vaccines are the most effective strategy to limit the extent of infection caused by SARS-CoV-2 virus, known as COVID-19. Vaccines represent a powerful strategy to reduce the morbidity associated with the disease. Nevertheless, alternative pharmacological protocols based on dexamethasone administration [1] and immunotherapy based on the used of interleukin-6 (IL-6) receptor agonists [2] should be taken into account as well.

Regarding vaccination, some studies have reported that vaccines encoding the SARS-CoV S protein elicited potent cellular and humoral immune responses in rodent models and in clinical trials [3,4]. In this context, the spike (S) protein of coronaviruses is crucial in SARS-CoV-2 infection. For this reason, S protein of SARS-CoV-2 has been identified as the most suitable target for vaccine development to trigger virus-specific T-cell responses as well as humoral immune responses [5]. The use of the S protein as an antigen provides protective antibodies that neutralise the virus and prevent infection, leading to an optimal cellular response [6,7]. The S protein contains two subunits, S1 and S2, which mediate receptor binding and membrane fusion, respectively. The S1 subunit contains a sequence called the receptor-binding domain (RBD) that is able to bind to angiotensin-converting enzyme-2 (ACE-2) receptor [8]. This particular domain used as an antigen is able to induce neutralizing antibodies and T-cell immune responses [9,10]. On the other hand, the administration of a single dose of adenovirus type 5-vectored COVID-19 vaccine in healthy individuals is safe and well tolerated [11]. This vaccine variant produces specific antiviral T-cell and humoral immune responses after 2 weeks of administration. In addition, a marked increase in the production of interferon-γ (IFN-γ), tumour necrosis factor-α (TNF-α) and interleukin-2 (IL-2) by CD4+ and CD8+ T cells post-vaccination has been observed [11].

In any case, the mechanism of COVID-19 vaccines could be summarized in three aspects: (a) mRNA vaccines that instruct the cells of the recipient to synthesize a harmless protein exclusive of COVID-19; (b) vaccines containing subunits or harmless portions of SARS-CoV-2 proteins; and (c) vector vaccines containing a modified version of a virus different of COVID-19. So far, the data suggest that all these vaccines offer protection against the disease, decreasing morbidity and mortality [12,13].

SARS-CoV-2 infection is a heterogeneous process, depending on age, sex, ethnicity, and comorbidities, among others. The dynamics of transmission of the virus and its contagion capacity remains to be established due to the appearance of new mutations. Therefore, avoiding transmission between patients is not the only strategy. To increase the effectiveness of vaccines should also be taken into account as a complementary safe alternative [14].

The evaluation of the efficiency and effectiveness of COVID-19 vaccines is complex, due to the lack of knowledge about the pathophysiology of the disease, and the diversity of vaccine formulations and strategies. The efficiency of a vaccine can be assessed based on the reduction of the infection percentage among vaccinated people compared to the frequency of infection among those who were not vaccinated, assuming that the vaccine is the cause of infection reduction. However, effectiveness represents the general health benefits provided by a vaccination program in the population. However, good efficiency does not always mean good effectiveness [15].

The use of adjuvant strategies can increase the effectiveness of vaccines against SARS-CoV-2, as observed in other vaccination protocols against other infectious diseases. In this context, one of the candidate nutrients is vitamin D, which has been shown to have a protecting effect against some viral infections, including dengue, influenza, and common cold [16,17]. The possibility has also been raised that vitamin D may prevent and/or mitigate COVID-19 infection [18]. Alternative compounds with immunomodulatory properties, such as glucans, have also been proposed as adjuvants [19].

The evidence regarding vitamin D effect in infectious and inflammatory processes, allow us to propose that vitamin D supplementation could help in the defense process induced by vaccines. This could be achieved by improving the production of antibodies against SAR-CoV-2 and/or improving the immune response against viral antigens, modulating the response of cytokines and inflammatory messengers. Therefore, in this review we propose to analyze and give a molecular explanation to a candidate role of vitamin D as adjuvant in vaccination against SARS-CoV-2.

## 2. Immune Response against SARS-CoV-2

Coronaviruses activate a battery of innate immune cells, including neutrophils, monocytes/macrophages, natural killer cells (NKCs), T cells, mast cells, resident endothelial and epithelial cells. This wide activation induces a particular response known as “cytokine storm” [20,21,22].

The first response after SARS-CoV-2 infection is carried out by innate immune cells, mainly mast cells, commonly located in the nasal epithelium and lower respiratory tract [23]. The disease severity relies on the ability of innate immune cells to stop or mitigate the infection [24]. Therefore, protective immunity against SARS-CoV-2 lies in humoral and cellular responses (acquired or adaptive immunity) [23]. The production of neutralizing antibodies against infection or after vaccine administration gives rise to the proliferation of memory-specific T cells (CD4+ and CD8+) against SARS-CoV-2 [23].

Adaptive immune responses occur after the innate immune response, participating as well in the elimination of the virus immediately and in subsequent infections. In this context, T cells display the ability to destroy viruses, but need the participation of antigen-presenting cells (monocytes/macrophages, dendritic cells, neutrophil B cells, and mast cells). These cells can transfer antigens to memory T cells as well as to B lymphocytes [25]. However, many of the processes resulting from SARS-CoV-2 infection remain to be characterized in more detail, including the mechanisms operating in antigen presentation, the exact innate cellular and humoral responses, and the characterization in more detail of the above-mentioned cytokine storm [26].

Increased interleukin-6 levels in COVID-19 patients may induce differentiation of Th17 cells and stimulate cytokine storm, inflammation, and pulmonary dysfunction [27,28]. Th17 cells produce interleukin-17 members. However, this cytokine family is also secreted by other immune cells, including neutrophils, mast cells, NKCs, and innate lymphocytes [29,30]. Both antibody production and increased number of specific memory T cells are instrumental to long-term immune protection against SARS-CoV-2 and to the prevention of severe forms of COVID-19. However, this does not imply that the two processes are going in the same direction. For example, convalescent individuals seem to present a vigorous memory T-cell response several months after SARS-CoV-2 infection, even in the absence of circulating specific antibodies against SARS-CoV-2 [31].

The humoral response includes antibodies against S protein and nucleocapsid protein (N). As mentioned before, S1 subunit of S protein contains the RBD that binds to ACE-2 receptor in human cells. Then, a rapid response appears with the production of virus-specific antibodies (IgG, IgA, and IgM) against different epitopes (antigenic determinants) of S protein [32]. It seems that the severity of the disease depends on IgG and IgA levels. The lowest levels seem to appear in the mildest disease cases [33,34,35]. On the other hand, antibodies against RBD appear before antibodies against the N protein that is associated to viral genetic material during infection [36]. For this reason, anti-RBD antibodies are more specific and sensible for diagnosis than anti-N. Seroconversion of RBDs in patients is rapid and occurs frequently, displaying low cross-reactivity with SARS-CoV-2 [10,37].

In this line, mast cells are innate immune cells participating in adaptive immunity processes, such as certain stress disorders, inflammatory pathologies, and virus infections. Mast cells are widely distributed in the body systems, acting against the attack of infectious microorganisms (viruses and bacteria) and neutralizing toxins [38]. In this context, mast cells function differently, exerting a protective role but at the same time causing harmful effects in the organism [39,40].

On the other hand, it has been reported that patients with a low concentration of serum vitamin D have elevated D-dimer (protein fragment that appears during blood clotting) levels. In addition, these patients display high B cell counts, reduced CD8+ T cells, uncertain clinical prognosis, and affected chest computed tomography [41]. Altogether, vitamin D appears to play a central role in innate and adaptive immunity, although its effect on adaptive immune responses is not fully understood. However, at least part of these effects depends on the suppression of auxiliary type 1 T cells (Th1) and the stimulation of suppressive regulatory T cells [42].

## 3. Inflammatory Response in COVID-19

During SARS-CoV2 infection, damaged cells induce inflammation in different systemic locations. This process is largely mediated by pro-inflammatory cells, such as granulocytes and macrophages, causing the described symptoms of fever, cough, fibrosis, and a large increase in pro-inflammatory cytokine levels. Cytokines are secreted glycoproteins that regulate different functions of immune cells, including apoptosis, proliferation, differentiation, activation, and maturation of lymphocytes and accessory cells [43,44]. In acute COVID-19 cases, an exaggerated inflammatory response, known as “cytokine storm,” has been described [22,28,45]. Finally, cytokines are also involved in the regulation of tissue distribution of leukocytes through the circulatory system [43,44].

Cytokines also modulate different functions in cells from various organs and body systems. Pleiotropy and redundancy in their functions are the main characteristics of these molecules. Cytokines include a wide family of extracellular messengers: interleukins, interferons, colony-stimulating factors, and many growth factors [43,44]. The main group of cytokines involved in leukocyte communication is the IL family. Other groups of cytokines modulate cell proliferation processes, such as colony-stimulating factors (e.g., granulocyte-macrophage colony-stimulating factor, and granulocyte colony-stimulating factor/G-CSF). Others regulate tumor cytotoxicity, such as TNFs. Finally, other cytokines inhibit viral replication, such as IFNs [43,44]. Thus, modulation of the immune system is instrumental in the prevention and therapy for inflammatory diseases [45,46].

The hyper-inflammatory response (“cytokine storm”) induced by SARS-CoV-2 needs to be taken into account [47], but it does not seem to be the central pathological abnormality in COVID-19 [48]. In the early stages of infection, pro-inflammatory cytokines, and chemokines (CCL) are produced, including IL-1β, IL-2, IL-6, IL-8, IFN-α, IFN-β, TNF-α, CCL2, CCL3, CCL5, and inducible protein-10 (IP-10) [49]. In the subsequent hyper-inflammatory phase, severe COVID-19 cases dramatically increase the levels of IL-2, IL-6, IL-7, IL-10, IP-10, CCL2 (also known as monocyte chemoattractant protein-1/MCP1), TNF-α, macrophage inflammatory protein-1α (MIP1α), and G-CSF. In particular, IL-6 and TNF-α display a wide range of fluctuation, in many cases exceeding the physiological range [49]. However, a lower production of these cytokines has been reported in patients with moderate SARS-CoV2 infection or in the early inflammatory stage. Thus, overproduction of cytokines and chemokines result in lung damage and life-threatening respiratory complications. This “cytokine storm” downregulates likely innate and adaptive immune responses against SARS-CoV-2 infection [50,51].

Low expression of IFN-γ related to a decrease of CD4+ and CD8+ T cells as well as NKCs, has been reported in cases of severe COVID-19 infection [52]. A high IL-6/IFN ratio seems to predict high severity in COVID-19 as well as lung damage as a result of “cytokine storm” [53]. Viral proteins, and in particular two non-structural polyproteins of SARS-CoV-2, inhibit the secretion and signaling of IFN- γ via Janus kinase (JAK) [54,55,56]. JAK blockade results in the decreased expression of the signal transducer and activator of transcription-1 (STAT1), a key transcription factor in the immune response [54].

The response of the immune system is directly modulated by the activity of the hypothalamus-pituitary-adrenal (HPA) axis. This interaction is established mainly through glucocorticoid secretion [43,44,57]. Stress subsequent to SARS-CoV-2 infection results in the release of the corticotrophin-releasing hormone (CRH) that triggers the HPA system [58]. In this context, Kempuraj et al. [59] indicate that mast cells respond to CRH and other neuropeptides that can initiate and exacerbate neuro-inflammation processes. The stress generated can diminish immune responses to infections compromising at the same time the response to vaccines [60].

## 4. Cytokine Response in Inflammation

Cytokines are extracellular glycoproteins produced mostly by leukocytes. These molecules modulate different processes in immune cells [43,44,57,61]. Cytokines are transiently secreted during the immune response. They bind with high affinity to specific membrane receptors to exert their effects in the target cells [43,44,57,61]. In this context, cytokines work as extracellular messengers in intercellular (paracrine) or systemic (endocrine) communication. Cytokines can stop their own synthesis by using autocrine, paracrine, or endocrine mechanisms, as well as the synthesis of other cytokines and their receptors. Inhibitory mechanisms comprise corticosteroid and eicosanoid synthesis, secretion of soluble receptors, and blockade of active signal transduction pathways [62].

Pleiotropy in their function is one the main characteristics of cytokines. In this context, IL-1, IL-6 and TNF-α participate in most inflammatory processes. For this reason, they are considered the main targets for therapeutic protocols [43,44,61,62,63,64,65]. Inflammation results from a cascade of molecular and cellular events that lead to fever and dilation of capillaries. These stressor-triggered responses are decisive for host response, activating defense mechanisms and recovery of tissue homeostasis. These processes require the elimination of altered molecules (i.e., oxidized proteins) and debris from damaged tissues [66,67]. However, the effects of cytokines on cells and target systems during an immune process are flexible, showing different levels of response, particularly for IL-1β, IL-6, and TNF-α [43,44,57,61,62,63,64,65].

In addition, IFN-α and IFN-β are involved in the immune response against the virus. In addition to its role in immunity, IFN-γ can modulate the inflammatory response as well. IFN-γ is produced by NKCs, working as a robust macrophage activator and inducing the nonspecific defense in host cells [68,69]. In this context, IFN-γ functions as a pro-inflammatory cytokine and increases the synthesis of additional inflammatory cytokines such as TNF-α. At the same time, IFN-γ regulates the expression of TNF-α receptors [70] and nitric oxide synthase (NOS) [71].

## 5. Vitamin D

Vitamin D has two main inactive precursors: Vitamin D2 (ergocalciferol) and Vitamin D3 (cholecalciferol). Vitamin D2 is synthesized by certain plants that are consumed through the diet. Vitamin D3 is synthesized in the skin after sun exposure. To be active, both forms need a double hydroxylation: one in the liver to form 25-(OH)D by the microsomal hydroxylase CYP2R1 for ergocalciferol and mitochondrial CYP27A1 for cholecalciferol, and the second hydroxylation in kidneys by the mitochondrial hydroxylase CYP27B1, yielding active 1,25-(OH)_2_D. Vitamin D is instrumental in the modulation of basic physiological processes, such as growth and development, bone formation and homeostasis, and nerve transmission. In addition, vitamin D modulates cell oxidative capacity, energy production, and immune function [72,73,74]. 1,25-(OH)_2_D functions as a ligand for vitamin D nuclear receptor (VDR), modulating the subsequent transcriptional events [75].

As an immunomodulatory micronutrient [18,19], vitamin D participates in the differentiation of monocytes to macrophages, increasing their chemotactic and phagocytic ability, preventing immunopathology and favoring efferocytosis [76]. In this context, it has been described that there is a connection between low vitamin D levels and increased risk of acute respiratory viral infections (ARVI) [77,78,79,80], leading to acute lung injury and respiratory distress syndrome [81]. Regarding other immune cell types such as dendritic cells, vitamin D diminish maturation and thereby, antigen presentation. In addition, vitamin D favors the expression of cathelicidin (an antimicrobial peptide), favoring microbial elimination [82]. Furthermore, vitamin D promotes a shift in T lymphocytes towards a Th2 profile and a Treg phenotype. This change may be explained by a direct action of vitamin D or an indirect pathway through vitamin D modulation of antigen-presenting cells [82].

### 5.1. Vitamin D against Infection

A relevant process in the immune system is the synthesis of the active form of vitamin D (1,25-(OH)_2_D) by antigen-presenting cells, including dendritic cells and macrophages [83]. The biosynthetic process seems to be modulated first, by the availability of 25-(OH)D, then by the induction of CYP27B1 hydroxylase by invading pathogens, and third by the transcriptional stimulation of target genes by the complex 1,25(OH)_2_D-VDR in the cells of the immune system.

In situations of vitamin D deficiency, the immune response would be affected, resulting in a drop of the innate immune function [84]. Therefore, vitamin D deficiency increases vulnerability to viral infections and the risk of subsequent periodic infections. For example, low serum vitamin D levels were associated with increased incidence of high-burden viral diseases, including influenza, hepatitis, AIDS (acquired immunodeficiency syndrome), and COVID-19 [85].

Therefore, the beneficial role of vitamin D against infection seems to be supported by three main mechanisms. As mentioned before, the first one refers to the improvement in the immune response [86]. The second refers to the barrier effect exerted through the stimulation of genes that encode proteins related to cell integrity and intercellular junctions such as occludin (tight junctions), connexin 43 (gap junctions), and cadherin E (adherent junctions). In this regard, many viruses alter the integrity of these barriers, increasing the degree of infectivity [87]. Therefore, the capacity of vitamin D in maintaining the entire cell barrier offers a complementary protection against infection. The third mechanism refers to the inhibition of the renin gene expression that affects the renin-angiotensin system (RAS), allowing to the activation of ACE-2 enzyme. ACE-2 is similar to ACE of RAS but with a different substrate specificity. The end product of ACE is angiotensin II, a potent vasoconstrictor peptide. Meanwhile, ACE-2 is a membrane protein expressed in all tissues, but with a particular high expression in lung epithelial cells. ACE-2 cleaves angiotensin II producing angiotensin 1–7, which is vasodilator and anti-inflammatory peptide due to its action on Mas receptor. The Mas pathway activated by angiotensin 1–7 binding supresses the activation of the extracellular signal regulated kinase/nuclear factro-κB (NF-κB) that plays a central role in inflammation [88,89]. As mentioned before, ACE-2 is the target of SARS-CoV-2.

### 5.2. Vitamin D and COVID-19

SARS-CoV-2 infection is associated with a wide variety of symptoms due to the inter-individual differences in immune responses. According to clinical reports, when the virus enters the body, it first goes through an incubation stage. During this phase, if the host immune system produces a specific immune response, the virus is eliminated, and the host escapes disease progression [90]. If the host immune system cannot eliminate the virus in this initial stage, the virus can attack tissues displaying expression of ACE-2 receptors, such as lungs and kidneys. The result is an activation of inflammatory processes mediated by the innate immune system (particularly macrophages and Th1 cells) with secretion of pro-inflammatory cytokines such as IL-1β and IL-18. This causes further organ damage and leads to the severe phase of the disease [90,91].

Since vitamin D plays a key role in immune system regulation, it can be assumed that vitamin D levels may correlate to the probability of SARS-CoV-2 infection [90,92]. In this context, it has been suggested that binding of vitamin D to VDR may decrease progression of COVID-19 [78] by decreasing cytokine/chemokine storm, modulating renin-angiotensin system, regulating neutrophil activation, maintaining the structure of the pulmonary epithelial barrier, stimulating repair processes in the epithelium, and reducing an increased risk of blood clotting [81,93,94,95] (Figure 1).

VDR is widely distributed in many tissues and works cooperating with other transcription factors. It also regulates expression of genes containing promoters with specific DNA sequences, called vitamin D response elements (VDRE) [75,96]. Due to the direct transcriptional activity, vitamin D-VDR complex modulates key physiological processes related to bone mineralization, detoxification actions, cell cycle events (including proliferation, differentiation, migration, and death), immune responses, as well as cellular metabolic processes [75,97].

Quesada et al. [98] suggested that administration of elevated doses of 25-(OH)D significantly reduced the stage of COVID-19 patients in emergency services. Low levels of circulating 25-(OH)D have been linked to an increased predisposition to SARS-CoV-2 infection [99] and severe progression of COVID-19 [100].

Finally, lymphocytes play an instrumental role in maintaining immune and inflammatory homeostasis, a key aspect to protect the body against viral infections [101]. In this context, the main part of the studies analyzed in the meta-analysis by Yang et al [102] noticed that SARS-CoV-2 infection decreases the number of lymphocytes in the most severe cases. However, this reduction is not evident in mild COVID-19 cases [103]. A possible relation between circulating levels of vitamin D and lymphocyte numbers remains to be investigated.

### 5.3. Vitamin D and the Inflammatory Response

Sport performance at high intensities is a well described inflammatory process. Many studies indicate that vitamin D deficiency results in deficient recovery, subsequently affecting sport performance [104]. On the other hand, elevated circulating levels of vitamin D are related to low injury rates, indirectly improving performance [105]. Vitamin D seems to decrease inflammation by inhibiting the production of pro-inflammatory cytokines. These include IL-6, which promotes the conversion of monocytes into macrophages, increasing at the same time the synthesis of further pro-inflammatory cytokines. Vitamin D action seems to be similar to the anti-inflammatory activity of the immune-modulator AM3. This drug reduces the production of pro-inflammatory cytokines, such as IFN-α, IL-2 and TNF-α [106,107,108,109]. Therefore, these sports studies have provided key information regarding the anti-inflammatory role of vitamin D in post-exercise recovery.

In a health context, suboptimal levels of vitamin D are linked to chronic diseases and damage in skeletal muscle. The muscular effects of vitamin D require binding to the intracellular VDR [107,110]. The VDR-vitamin D complex regulates the expression of hundreds of genes that code for proteins that accomplish key cellular functions. In this context, the VDR-vitamin D complex stimulates eNOS expression, regulating at long term the synthesis of NO [107,110,111,112,113]. Regulated production of NO can promote angiogenesis of endothelial cells [114,115]. In addition, it has been shown that the absence of the VDR gene decreases the bioavailability of L-arginine (the precursor of NO). This results in an increased expression of arginase-2, an enzyme that converts L-arginine in ornithine and urea [112].

On the other hand, optimal circulating levels of vitamin D are instrumental in maintaining an efficient anti-inflammatory response [116,117]. Therefore, supplementation with 1,25(OH)_2_D to restore adequate circulating levels, suppresses pro-inflammatory cytokine expression [118], allowing the VDR-vitamin D complex to stimulate the recovery of skeletal muscle function [119,120,121,122,123,124]. The mechanisms proposed to explain the action of vitamin D in the restoration of muscle strength are: (a) modulating gene expression by direct binding of 1,25(OH)_2_D to VDR in muscle cells [125,126,127], and (b) promoting the transport of calcium into the sarcoplasmic reticulum necessary for muscle contraction (only tested in animal models) [125].

In this context, low levels of vitamin D in circulation are linked to impaired muscle function, decreased muscle strength and disorders in muscle metabolism [128]. The evidence suggests that supplementation with correct levels of vitamin D restores performance and avoids muscle injuries, improving muscle strength [129]. At a subcellular level, vitamin D supplements improve mitochondrial function, evidenced by correct morphology, stimulated mitochondrial protein synthesis, and inhibition of free radical production [130]. Therefore, vitamin D deficiency favors oxidative stress resulting from mitochondrial dysfunction in skeletal muscle [131].

Therefore, vitamin D plays a key role in the control of inflammation. However, the modulation of the anti-inflammatory response remains to be investigated in the control and destruction of respiratory pathogens, including SARS-CoV-2 [132,133].

## 6. Why Vitamin D Could Act as Adjuvant in the Vaccination Protocol in COVID-19

Currently, vaccines are a better tool in the prevention of infectious diseases, such as COVID-19. Vaccines against SARS-CoV-2 present different types [12,13]: administration with a single dose, vaccination with two doses and vaccination with a third dose. Therefore, several vaccination protocols are under research. However, we think that it would be important to consider the possibility of using adjuvants to improve vaccine performance, independently of the dose number.

The role of cytokines should be considered instrumental in the vaccination process. As indicated before, cytokines regulate the distribution of circulating leukocytes and exert modulatory effects on target cells of various organs and body systems [134,135,136]. Certain cytokines produce cytotoxicity and inhibition of viral replication, such as TNFs and IFNs, respectively [134,135,136,137]. Therefore, cytokines play a key function in the immune response against viral infection [138]. In this context, vitamin D modulates the expression of pro-inflammatory cytokines. Downregulation of the adaptive immune response in patients with severe COVID-19 presents a marked decrease in CD4+, CD8+ and regulatory T cells, accelerating the production of pro-inflammatory cytokines [139,140].

In mice, vitamin D favors the migration of skin dendritic cells to Peyer plaques after antigen-induced maturation [141]. In this context, dendritic cells work as antigen-presenting cells for activation of immunity in mucosal tissues [141]. The immune-stimulant effects on the response to antigens in circulation and mucosal tissues with the administration of 1,25-(OH)_2_D have been described in pigs [142] and cattle [143]. However, recent reports on the use of 1,25-(OH)_2_D as an immune-modulator with a flu vaccine did not show an increase in antigen-specific immune response [144,145].

The immune-regulatory effects of vitamin D are primarily due to the activation of anti-inflammatory signaling pathways. In this context, vitamin D has been used as a beneficial adjuvant in vaccination against tuberculosis [146]. These authors indicate that vitamin D may exert a function in the immunity induced by vaccination against BCG (Bacillus Calmette-Guérin) contributing likely to the non-specific effects described after vaccination [146].

In addition, 1,25-(OH)_2_D has been administered to mice vaccinated with different vaccines, such as inactivated polio vaccine (IPV) [147], Hemophilus influenzae type b oligosaccharide conjugated with diphtheria toxoid vaccine [142], and hepatitis B surface antigen (HBsAg) vaccine [145]. In all these cases, the 1,25-(OH)_2_D administration stimulated the production of antigen-specific immunity in mucosal tissues (IgA and IgG antibodies), resulting in a greater systemic immune response [82,142,147,148].

## 7. Conclusions

Although the function of vitamin D in different body systems offers many possible interactions against the mechanisms by which the SARS-CoV-2 virus infects humans, the available data are still not conclusive. However, in view of the evidences presented in this review, we think that vitamin D may be a good adjuvant element in vaccination (Figure 2). Possible action mechanisms include attenuation of pro-inflammatory cytokine production and at the same time increasing the production of antibodies. Recent reports on circulating vitamin D levels in COVID-19 patients reveal a correlation between vitamin D insufficiency and the severity of respiratory disease caused by SARS-CoV-2. Higher COVID-19-related mortality has also been observed in countries with low average vitamin D circulating levels [149,150].

## Figures and Tables

**Figure 1 medicina-57-01321-f001:**
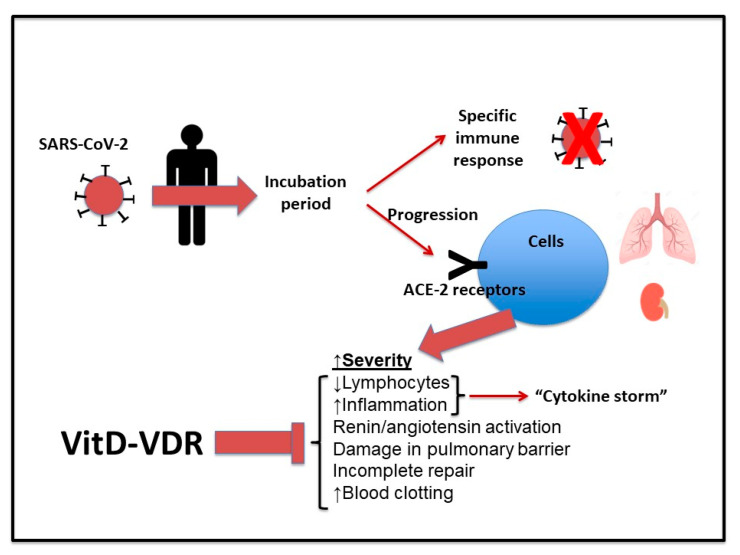
Scheme of the proposed role of vitamin D during SARS-CoV-2 infection. An incubation phase occurs when the virus enters the body. If the host produces a specific immune response, the virus is eliminated. If the host immune system cannot eliminate the virus, this can infect tissues with ACE-2 receptors (lungs and kidneys). The result is an increase in the severity of the disease. Vitamin D-VDR complex may attenuate the symptoms decreasing progression of COVID-19. (↑) Indicates activated process/increased number (↓) Decreases.

**Figure 2 medicina-57-01321-f002:**
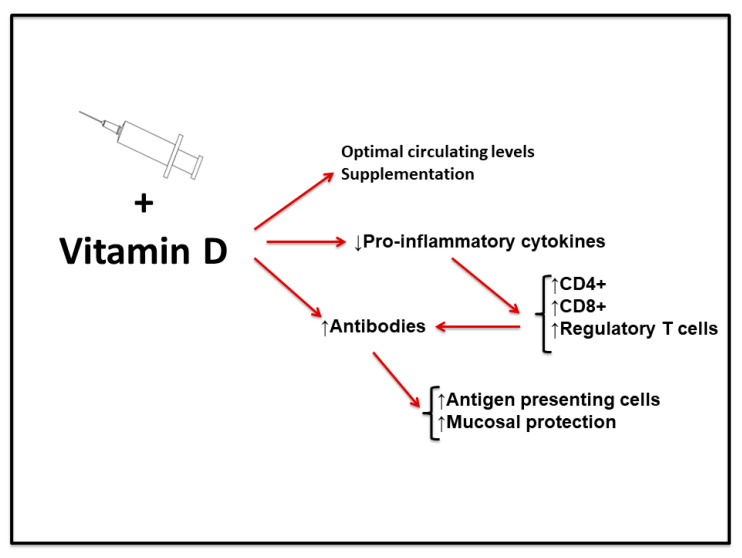
Scheme of the proposed adjuvant effect of vitamin D for COVID-19 vaccination. Optimal circulating levels or supplementation with vitamin D may work as an adjuvant for COVID vaccination. Proposed mechanisms are attenuation of pro-inflammatory cytokine production and increasing the production of antibodies. (↑) Indicates activated process/increased number and (↓) indicates decreased production.

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
