# Peer review of "The Immunomodulatory Function of Vitamin D, with Particular Reference to SARS-CoV-2"

_medicina, 2021, doi:10.3390/medicina57121321_

Round 1

Reviewer 1 Report

An interesting article proposing vitamin D as an adjuvant in the vaccination against Covid 19:

I think that, although Covid 19 has been fully discussed, too little has been reported about the immune response to covid vaccines; this part should be expanded.

Page 2 line 79 "The disease severity relies on the ability of innate immune cells to stop or mitigate the infection." this sentence needs a reference, such as: doi: 10.1111/dth.13681. 

page 3 line 88-91 "However, many of these processes resulting from SARS-CoV-2 infection remain to be characterized in more detail, including the mechanisms operating in antigen presentation, the exact innate cellular, and humoral responses, and the characterization in more detail of the above-mentioned cytokine storm." this paragraph needs some reference, such as: doi: 10.3390/medicina57080828.

Author Response

REVIEWER-1

An interesting article proposing vitamin D as an adjuvant in the vaccination against Covid 19:

I think that, although Covid 19 has been fully discussed, too little has been reported about the immune response to covid vaccines; this part should be expanded.

ANSWER: This an important point, although the accumulated knowledge is scarce at present. In any case, we have addressed this point in the second paragraph of Introduction.

Page 2 line 79 "The disease severity relies on the ability of innate immune cells to stop or mitigate the infection." this sentence needs a reference, such as: doi: 10.1111/dth.13681.

ANSWER: The reference (Reference 24) suggested by the Reviewer has been added (See Line 100). 

Page 3 line 88-91 "However, many of these processes resulting from SARS-CoV-2 infection remain to be characterized in more detail, including the mechanisms operating in antigen presentation, the exact innate cellular, and humoral responses, and the characterization in more detail of the above-mentioned cytokine storm." this paragraph needs some reference, such as: doi: 10.3390/medicina57080828.

ANSWER: The reference (Reference 26) suggested by the Reviewer has been added (See Line 112). 

Reviewer 2 Report

The authors present data regarding an interesting proposal that vitamin D should be used as an adjunct to improve the efficacy of vaccines to SARS-CoV-2. This is clearly a timely piece of work. I found the paper of interest but I expected more detail to justify the suggestion. Please see my comments below, with particular emphasis on my final paragraph.

One of the main issues I have is that the manuscript needs significant correction to the grammar used eg: Ln 36: “The performance of COVID-19 vaccines could be summarized in 3 aspects:”. The use of “performance” here is not quite correct, “mechanism” would be a better choice.

Eg Ln 45-47: “To increase the effectiveness of vaccines should also be taken into account as a complementary effective alternative [3]”. This is clunky. There are multiple other examples. I sympathize with the fact English may not be their first language and that I would be unable to write anything in Spanish. Nonetheless, it requires review from the grammar perspective.

Ln 35 “Actually, this is the only way to reduce the morbidity associated to the disease.” I think this fails to acknowledge the benefit of dexamethasone in those on Respiratory support (see Dexamethasone in Hospitalized Patients with Covid-19. The RECOVERY Collaborative Group*. The RECOVERY  February 25, 2021, N Engl J Med 2021; 384:693-704. DOI: 10.1056/NEJMoa2021436). It also fails to consider recent research using IL-6 receptor antagonists/immunotherapy.

Ln 75-the authors suggest a “cytokine storm” is a key factor (this is repeated at Ln 138 and Ln 152 and throughout the manuscript). I can understand why they have put that as others have suggested the same, however a recent excellent review (Osuchowski MF Winkler MS Skirecki T et al. The COVID-19 puzzle: deciphering pathophysiology and phenotypes of a new disease entity. Lancet Respir Med. 2021; https://doi.org/10.1016/S2213-2600(21)00218-6) concludes that although systemic inflammation is clearly important, available data do not indicate that a so-called cytokine storm is the central pathological abnormality in COVID-19. I therefore think reference to a cytokine “storm” should be toned down.

Ln 173-188. While I agree COVID-19 infection can stimulate the HPA axis and increase CRH and cortisol as part of a stress response I don’t accept that “HPA activation caused by CRH release due to SARS-CoV-2 infection leads to anxiety, stress-induced depression, psychiatric disorders and post- traumatic stress”. I strongly believe that the anxiety, depression and PTSD are related much more to the psychosocial factors associated with acute severe illness in general rather than being specifically due to CRH release. I do not accept that this is “cause and effect”. “In this context, stressful situations arising from COVID-19 undoubtedly affect the immune response, physical and mental health, favoring situations of drug abuse”. I also find this an over-simplification and I am uncomfortable with reference to drug abuse.

Ln 279-282: “Quesada et al [88] suggested that administration of elevated doses of 25-(OH)D significantly reduced the stage of COVID-19 patients in emergency services. Therefore, low levels of circulating 25-(OH)D have been linked to an increased predisposition to SARS-CoV-2 infection [89] and severe progression of COVID-19 [90].” The way this is written (specifically the use of “Therefore”) suggests that the evidence from ref 88 justifies the evidence from ref 89. This could be very misleading and “Therefore” needs to be removed.

Ln 303 “Lesion”. Do they mean “injury”?

In Section 6, Ln 368-371 where they make the claim for Vitamin D as an adjuvant in vaccination programmes they start to talk about vitamin D and a role in recovery, which is inappropriate here.

My main issue is that for a paper purporting to make the case for adjuvant vitamin D to enhance vaccine response while they present a lot of interesting data about vitamin D and it’s role in immunomodulation the data where it has actually been given alongside vaccines in mice is given only brief discussion (Ln 375-380). Indeed, while they present a lot of interesting data for the immune association of vitamin D the case they make for it as an adjunct in vaccination is not articulated strongly. I don’t think the manuscript detail supports the title: “Could be Vitamin D an Adjuvant to Enhance the Efficacy of SARS-CoV-2 Vaccines?” This is a shame because it is an enticing and attractive hypothesis. Perhaps if they were to re-title the paper “The immunomodulatory function of vitamin D, with particular reference to SARS-CoV-2” or something similar. They could then keep the bulk of the text and introduce their idea of it as an adjunct to vaccination.

Author Response

REVIEWER-2

The authors present data regarding an interesting proposal that vitamin D should be used as an adjunct to improve the efficacy of vaccines to SARS-CoV-2. This is clearly a timely piece of work. I found the paper of interest but I expected more detail to justify the suggestion. Please see my comments below, with particular emphasis on my final paragraph.

One of the main issues I have is that the manuscript needs significant correction to the grammar used eg: Ln 36: “The performance of COVID-19 vaccines could be summarized in 3 aspects:”. The use of “performance” here is not quite correct, “mechanism” would be a better choice.

ANSWER: We are not native English speakers. Following Reviewer suggestions, we plan to use the English editing service offered by the editorial to correct all grammar errors, if the manuscript is accepted for publication. In any case, the mistake in Line 36 (now Line 57) has been corrected accordingly.

Eg Ln 45-47: “To increase the effectiveness of vaccines should also be taken into account as a complementary effective alternative [3]”. This is clunky. There are multiple other examples. I sympathize with the fact English may not be their first language and that I would be unable to write anything in Spanish. Nonetheless, it requires review from the grammar perspective.

ANSWER: We insist in the use of the English editing service available from the editorial, but we need to have a final version to do the final correction. In any case, the clunky sentence has been corrected accordingly (see Line 67).

Ln 35 “Actually, this is the only way to reduce the morbidity associated to the disease.” I think this fails to acknowledge the benefit of dexamethasone in those on Respiratory support (see Dexamethasone in Hospitalized Patients with Covid-19. The RECOVERY Collaborative Group*. The RECOVERY  February 25, 2021, N Engl J Med 2021; 384:693-704. DOI: 10.1056/NEJMoa2021436). It also fails to consider recent research using IL-6 receptor antagonists/immunotherapy.

ANSWER: We know the benefits of the pharmacological treatment with dexamethasone and the promising immunotherapy using IL-6 receptor agonists. We wanted to focus in the use of vaccines and this is the reason why we have not mentioned the corresponding references to alternative therapy protocols. In any case, it is a very good idea to bring this information briefly to the reader (See Lines 36-39).  

Ln 75-the authors suggest a “cytokine storm” is a key factor (this is repeated at Ln 138 and Ln 152 and throughout the manuscript). I can understand why they have put that as others have suggested the same, however a recent excellent review (Osuchowski MF Winkler MS Skirecki T et al. The COVID-19 puzzle: deciphering pathophysiology and phenotypes of a new disease entity. Lancet Respir Med. 2021; https://doi.org/10.1016/S2213-2600(21)00218-6) concludes that although systemic inflammation is clearly important, available data do not indicate that a so-called cytokine storm is the central pathological abnormality in COVID-19. I therefore think reference to a cytokine “storm” should be toned down.

ANSWER: According to this new report, we have “toned down” the reference to the “cytokine storm” (See Lines 170-171).

Ln 173-188. While I agree COVID-19 infection can stimulate the HPA axis and increase CRH and cortisol as part of a stress response I don’t accept that “HPA activation caused by CRH release due to SARS-CoV-2 infection leads to anxiety, stress-induced depression, psychiatric disorders and post- traumatic stress”. I strongly believe that the anxiety, depression and PTSD are related much more to the psychosocial factors associated with acute severe illness in general rather than being specifically due to CRH release. I do not accept that this is “cause and effect”. “In this context, stressful situations arising from COVID-19 undoubtedly affect the immune response, physical and mental health, favoring situations of drug abuse”. I also find this an over-simplification and I am uncomfortable with reference to drug abuse.

ANSWER: The psychological aspects related to COVID-19 have been eliminated according to Reviewer suggestions. See last paragraph of Section 3.

Ln 279-282: “Quesada et al [88] suggested that administration of elevated doses of 25-(OH)D significantly reduced the stage of COVID-19 patients in emergency services. Therefore, low levels of circulating 25-(OH)D have been linked to an increased predisposition to SARS-CoV-2 infection [89] and severe progression of COVID-19 [90].” The way this is written (specifically the use of “Therefore”) suggests that the evidence from ref 88 justifies the evidence from ref 89. This could be very misleading and “Therefore” needs to be removed.

ANSWER: “Therefore” has been eliminated according to Reviewer suggestion (See Line 307).

Ln 303 “Lesion”. Do they mean “injury”?

ANSWER: We changed “lesion” by “injury” accordingly (See Line 330).

In Section 6, Ln 368-371 where they make the claim for Vitamin D as an adjuvant in vaccination programmes they start to talk about vitamin D and a role in recovery, which is inappropriate here.

ANSWER: The reference regarding a role of vitamin D in recovery has been eliminated as suggested by the Reviewer. See paragraph 4 of Section 6.

My main issue is that for a paper purporting to make the case for adjuvant vitamin D to enhance vaccine response while they present a lot of interesting data about vitamin D and it’s role in immunomodulation the data where it has actually been given alongside vaccines in mice is given only brief discussion (Ln 375-380). Indeed, while they present a lot of interesting data for the immune association of vitamin D the case they make for it as an adjunct in vaccination is not articulated strongly. I don’t think the manuscript detail supports the title: “Could be Vitamin D an Adjuvant to Enhance the Efficacy of SARS-CoV-2 Vaccines?” This is a shame because it is an enticing and attractive hypothesis. Perhaps if they were to re-title the paper “The immunomodulatory function of vitamin D, with particular reference to SARS-CoV-2” or something similar. They could then keep the bulk of the text and introduce their idea of it as an adjunct to vaccination.

ANSWER: We changed the title according to Reviewer suggestion. The new title is less tempting for the reader. In this way, we introduce our hypothesis of vitamin D as adjuvant in the text accordingly.

Reviewer 3 Report

The article is excelent. Two aspects could be added to the text: the barrier effect and the inhibition or RAS. Vitamin D can be beneficial by two different mechanisms: 1) the improvement of the immune system due to everything that has been clearly explained in the text, but also by a barrier effect that would be exerted through the stimulation of genes that encode proteins related to cell integrity and junctions such as occludin (tight junctions), connexin 43 (gap junctions) and cadherin E (adherent junctions). In this regard, it should be clarified that, in general, viruses alter the integrity of these barriers, which increases their degree of infectivity, therefore the activity of maintaining the entire cell barrier demonstrated for vitamin D is promising. 2) Inhibiting the Renin Angiotensin system (RAS ) for inhibiting the Renin gene, and also for intervening in the fate of Angiotensin II, stimulating the ACE2 enzyme, producing more Angiotensin 1-7, which is vasodilator and anti-inflammatory due to its action on its MAS receptor. In this way, there is less action of Angiotensin II on its AT1R receptor.

Articles that can be added are: 

Schwalfenberg GK. A review of the critical role of vitamin D in the functioning of the immune system and the clinical implications of vitamin D deficiency. Mol Nutr Food Res 2011;55:96-108.

 Kast JI, McFarlane AJ, Globinska A, Sokolowska M, Wawrzyniak P, Sanak M, Schwarze J, Akdis CA, Wanke K. Respiratory syncytial virus infection influences tight junction integrity. Clin Exp Immunol 2017;190:351-359.

Diaz-Curiel M, Cabello A, Arboiro-Pinel R, Mansur JL, Heili-Frades S, Mahillo-Fernandez I, Herrero-González A, Andrade-Poveda M. The relationship between 25(OH) vitamin D levels and COVID-19 onset and disease course in Spanish patients. J Steroid Biochem Mol Biol. 2021 Sep;212:105928.

Author Response

REVIEWER-3

The article is excelent. Two aspects could be added to the text: the barrier effect and the inhibition or RAS. Vitamin D can be beneficial by two different mechanisms: 1) the improvement of the immune system due to everything that has been clearly explained in the text, but also by a barrier effect that would be exerted through the stimulation of genes that encode proteins related to cell integrity and junctions such as occludin (tight junctions), connexin 43 (gap junctions) and cadherin E (adherent junctions). In this regard, it should be clarified that, in general, viruses alter the integrity of these barriers, which increases their degree of infectivity, therefore the activity of maintaining the entire cell barrier demonstrated for vitamin D is promising. 2) Inhibiting the Renin Angiotensin system (RAS) for inhibiting the Renin gene, and also for intervening in the fate of Angiotensin II, stimulating the ACE2 enzyme, producing more Angiotensin 1-7, which is vasodilator and anti-inflammatory due to its action on its MAS receptor. In this way, there is less action of Angiotensin II on its AT1R receptor.

Articles that can be added are: 

Schwalfenberg GK. A review of the critical role of vitamin D in the functioning of the immune system and the clinical implications of vitamin D deficiency. Mol Nutr Food Res 2011;55:96-108.

Kast JI, McFarlane AJ, Globinska A, Sokolowska M, Wawrzyniak P, Sanak M, Schwarze J, Akdis CA, Wanke K. Respiratory syncytial virus infection influences tight junction integrity. Clin Exp Immunol 2017;190:351-359.

Diaz-Curiel M, Cabello A, Arboiro-Pinel R, Mansur JL, Heili-Frades S, Mahillo-Fernandez I, Herrero-González A, Andrade-Poveda M. The relationship between 25(OH) vitamin D levels and COVID-19 onset and disease course in Spanish patients. J Steroid Biochem Mol Biol. 2021 Sep;212:105928.

ANSWER: Suggestions made by the Reviewer have been included. See the last paragraph of Section 5.1.